# Helios: Learning and Adaptation of Matching Rules for Continual In-Network Malicious Traffic Detection

## Abstract

Network Intrusion Detection Systems (NIDS) are critical for web security by identifying and blocking malicious traffic. In-network NIDS leverage programmable switches for high-speed traffic processing. However, they are unable to reconcile the fine-grained classification of known classes and the identification of unseen attacks. Moreover, they lack support for incremental updates. In this paper, we propose Helios, an in-network malicious traffic detection system, for continual adaptation in attack-incremental scenarios. First, we design a novel Supervised Mixture Prototypical Learning (SMPL) method combined with clustering initialization to learn prototypes that encapsulate the knowledge, based on the weighted infinity norm distance. SMPL enables known class classification and unseen attack identification through similarity comparison between prototypes and samples. Then, we design boundary calibration and overlap refinement to transform learned prototypes into priority-guided matching rules, ensuring precise and efficient in-network deployment. Additionally, Helios supports incremental prototype learning and rule updates, achieving low-cost hardware reconfiguration. We implement Helios on a Tofino switch and evaluation on three datasets shows that Helios achieves superior performance in classifying known classes (92%+ in ACC and F1) as well as identifying unseen attacks (62% - 98% in TPR). Helios has also reduced resource consumption and reconfiguration time, demonstrating its scalability and efficiency for real-world deployment.

## CCS Concepts

• **Security and privacy** → **Network security**.

## Keywords

Malicious traffic detection, Programmable switches, Prototypical learning

## 1 INTRODUCTION

Network Intrusion Detection Systems (NIDS) are essential for securing web services, as they classify malicious traffic from mixed user traffic to preserve network integrity. However, traditional NIDS, such as those based on Deep Packet Inspection (DPI), overly rely on experts to manually mine the attack fingerprints and struggle to accurately detect increasingly sophisticated cyber threats [5, 31]. Recently, deep learning (DL)-based supervised [40] and unsupervised [16, 21] methods have emerged, leveraging the powerful feature extraction capabilities of neural networks to uncover hidden patterns in network traffic for accurate attack identification. However, their high computational complexity necessitates uploading traffic from the network environment to the control plane (e.g., x86 servers equipped with GPUs) for processing, resulting in significant network bandwidth consumption and processing delays that hinder timely attack detection and can lead to financial losses [8, 26],

particularly in large-scale data centers. Furthermore, while an increase in model parameters improves classification accuracy, it also increases training time and restricts efficient updates.

In response to the increasing demand for real-time and high-speed traffic processing, the in-network traffic classification paradigm has emerged [25, 36, 38, 42]. Unlike DL-based NIDS, which are typically deployed on GPUs, the in-network paradigm utilizes programmable switches to directly perform inference within the data plane. This deployment enables terabit-per-second (Tbps) throughput while maintaining nanosecond-level latency. However, programmable switches rely on match-action logic and support only simple instructions, such as integer addition and bit shifts, making it infeasible to deploy DL-based models which heavily require floating-point operations [16, 40]. To overcome the computational limitations of programmable switches, for example, [1, 10, 38, 41, 42] transform tree-based machine learning models into match-action rules, while [22, 23] directly extract rules from data. In [36, 37], knowledge distillation is used to distill neural networks into a lightweight binary decision tree (BDT). Additionally, [39] achieves a conversion between regular expressions (RE) and byte-level recurrent neural networks (BRNN).

However, most existing in-network methods lack consideration of continual model updates. They may fail to identify unseen attacks because they completely partition the feature space based on existing classes, leaving no space for unseen attack types. Moreover, they require full retraining to update the model. In real network scenarios, continual updating and maintenance of NIDS after its initial deployment are essential as zero-day attacks emerge continuously [2, 32]. This attack-incremental nature of networks necessitates that NIDS refine its countermeasures over time. While some latest unsupervised methods [10, 22, 23] can identify previously unseen attacks, and [23] supports incremental model updates, they are limited to binary classification tasks and fail to meet the fine-grained classification demands of NIDS [40].

In this paper, we aim to design a new in-network NIDS that meets the following requirements: **1) proficient classification ability**: The system should perform robust classification on both known and new attacks. It should achieve high accuracy in multi-class classification for known classes while effectively identifying unseen attacks without prior knowledge. **2) scalable hardware deployment**: Given the limited computational resources and memory of programmable switches, the system should efficiently manage hardware resources to achieve high-throughput traffic processing without incurring excessive overhead. **3) incremental model updates**: The system should support incremental updates without requiring full retraining when adapting to new classes (i.e., administrator's changing requirements like newly detected attacks). This reduces model retraining time and minimizes switch interruption caused by reconfiguration, which is critical in high-speed environments (e.g., large web service provider networks serving millions of

users). Despite advancements, none of the current state-of-the-art in-network solutions meet all of these requirements simultaneously.

We propose Helios[1], a framework designed for continual in-network malicious traffic detection in attack-incremental scenarios. Helios consists of three key modules: *Attack Knowledge Prototyping*, *Priority-Guided Rule Transformation* and *Continual Rule Adaptation*.

*Attack Knowledge Prototyping* learns a set of prototypes that encapsulate the knowledge of classes (including benign and known attacks). We propose *Supervised Mixture Prototypical Learning (SMPL)*, based on Supervised Prototypical Learning (SPL) [6] and incorporating clustering-based initialization to achieve precise classification. Unlike most SPL methods [11, 27] that rely on complex transformations incompatible with programmable switches and compact each class into a single prototype, Helios operates directly on raw features and assigns multiple prototypes to each class in order to effectively capture diverse patterns. Specifically, Helios uses the weighted infinity norm to measure similarity, which aligns with the matching capabilities of programmable switches.

We design the *Priority-Guided Rule Transformation* method to convert the learned prototypes into range-based matching rules after completing prototype training, enabling efficient deployment on switches for high-speed packet processing. Specifically, Helios calibrates the acceptance boundaries of prototypes to enhance generalization ability. Then, Helios computes and assigns priorities to the existing prototype-transformed rules, while also introducing higher-priority rules for extended coverage, thereby ensuring optimal classification results in all overlapping regions.

During operations, when a new attack appears and network administrators collect and label data samples for the new attack, the *Continual Rule Adaptation* module performs incremental rule updates. We retain existing rules that are not matched by any new attack samples and then conduct incremental prototype learning on the updated training set. The newly learned rules are incrementally incorporated, ensuring low-cost hardware reconfiguration.

We implement Helios using the P4$_{16}$ [9] language and deploy it on a real Tofino switch for hardware evaluation, achieving a packet processing of 100Gbps per port. To evaluate the performance of Helios, we conduct experiments across three public attack traffic datasets [3, 28, 29]. The experimental results show that Helios achieves high accuracy for known classes (95.78%) while effectively identifying unseen attacks (98.21%). Furthermore, Helios achieves reduced switch resource consumption and less table entry reconfiguration time compared to baselines [37, 38, 41, 42] (up to 4.5x faster), demonstrating its scalability and efficiency for real deployment.

In summary, we make the following contributions:

- We propose Helios, the first in-network solution for continual malicious traffic detection in attack-incremental scenarios, capable of accurately identifying known classes and unseen attacks.
- We develop an innovative learning method that integrates *Supervised Mixture Prototypical Learning (SMPL)* with clustering initialization, specifically designed to enhance performance.
- We develop a *Priority-Guided Rule Transformation* method to resolve overlapping between rules. We also design an incremental update mechanism to enable efficient switch reconfiguration.

- We prototype Helios and perform comprehensive experiments to demonstrate its performance[2].

## 2 BACKGROUND

### 2.1 Malicious Traffic Detection and Challenges

Network Intrusion Detection Systems (NIDS) are essential for detecting malicious activities and anomalies. Traditional NIDS, such as those based on feature engineering or deep packet inspection (DPI), rely heavily on expert knowledge, limiting their adaptability to evolving threats. The advent of deep learning-based NIDS has significantly improved detection accuracy but at the cost of higher computational complexity and latency. For instance, even real-time NIDS [15] are constrained to throughput rates around 10 Gbps, far below the 100+ Gbps requirements of large-scale cloud and web service networks.

Programmable switches, built on the P4 language [4] and Protocol Independent Switch Architecture (PISA), offer a solution for high-speed packet processing directly within the data plane. They support custom table-based match-action pipelines deployed on switch ASICs, enabling network functions to operate at line rate. However, programmable switches are limited in computational capability, supporting only basic operations like integer addition and bit shifts, without support for loops or floating-point computations. This makes it challenging to deploy DL-based NIDS directly on switches, as they often require complex calculations and logic to effectively detect attacks [16, 40].

Existing methods address the challenges through two primary approaches. The first focuses on directly deploying tree-based models by converting them into rules that programmable switches can execute. For example, IIsy [38] introduces a mapping strategy to offload a decision tree (DT), while NetBeacon [42] offloads random forests (RF) by combining feature encoding and decision tables. Similarly, Flowrest [1] implements flow-level inference using RF. The second aims to develop models that are inherently suitable to the computational limitations of programmable switches by incorporating advanced machine learning techniques. For example, Mousika [36] leverages knowledge distillation to train the ternary matching-based binary decision tree (BDT) with the assistance of neural networks, enabling lightweight resource usage on switch. Metis [39] transforms regular expressions (RE) into trainable byte-level recurrent neural networks (BRNN), preserving domain-specific expert knowledge while allowing supervised optimization.

These methodologies mark notable progress in in-network attack detection, enabling precise and high-speed traffic processing. However, they still face several challenges, as detailed below.

**C1: proficient classification ability.** As highlighted in [40], an ideal model should be capable of both multi-class classification of known classes and identification of unknown attacks. This provides fine-grained classification results that enable administrators to take more targeted countermeasures and enhance system reliability by ensuring robustness against zero-day attacks. However, most existing supervised solutions [36–39, 41, 42] assume a fixed set of attack types, with the expectation that traffic data for all attack classes is available in advance. As demonstrated in section 5.2,

---

[1]Helios is a fictional Greek god, known for his ability to illuminate the world with his light, uncovering hidden dangers and revealing them earlier than others.

[2]We will open source the code.

despite our best efforts to extend these models, they still struggle to effectively detect unseen attacks. Furthermore, while unsupervised methods [10, 22, 23] can identify unseen attacks, they are limited to binary classification tasks and lack support for fine-grained malicious detection.

**C2: scalable hardware deployment.** To achieve high throughput packet processing, the model should enable in-network deployment. While some recent NIDS [21, 40] can detect unseen attacks, they rely heavily on floating-point operations and complex logical computations, making them impractical for deployment on resource-constrained network devices (e.g., programmable switches). Additionally, some rule-learning methods [30, 34] based on interpretability have been proposed, but they still require weighted probability adjustments after rule matching, which hinders their deployment. Additionally, the deployment should reduce hardware resource usage to preserve capacity for other essential network functions (e.g., routing). Although Metis [39] can be deployed on the switch, it consumes excessive hardware resources, occupying nearly all pipeline stages (i.e., 11 out of 12) even for binary classification tasks, thus restricting its scalability.

**C3: incremental model update.** The model should support incremental updates and efficient hardware reconfiguration to reduce retraining costs and avoid disruption to the attack detection. Most existing methods require full model retraining (i.e., learning from scratch), which necessitates updating all hardware table entries. This is particularly problematic in high-speed networks, where even a brief interruption can affect large volumes of traffic. Among in-network methods, only Genos [23] supports incremental updates. However, as mentioned earlier, it is limited to binary classification.

In summary, state-of-the-art NIDS fail to address all of the aforementioned challenges. Therefore, we propose Helios, which leverages learning techniques such as Supervised Prototypical Learning (SPL) and boosting to enable continual in-network malicious traffic detection.

## 2.2 Learning Techniques

**Supervised Prototypical learning.** In prototypical learning [6], a sample is classified into the class of the prototype to which it is most similar in the hidden space. If a sample shows little similarity to any prototype, it is classified as an out-of-distribution (OOD) sample, and therefore considered as belonging to an unseen class. Existing methods [11, 27] typically assign a single prototype to each class, which works well because the features are processed by neural networks, allowing a single prototype to effectively differentiate between classes with minimal overlap in acceptance ranges.

However, in the context of the data plane, which does not support feature processing (e.g., linear or nonlinear transformations), prototypes need to directly compare with original features. Consequently, we assign multiple prototypes to each class to enhance representation capability. However, this introduces new challenges, as the acceptance boundaries for prototypes become difficult to define, and an increase in the number of prototypes can lead to considerable overlaps. We address these issues in Section 4.2.

**Boosting.** Boosting is a powerful learning technique commonly used in machine learning [7, 13, 14] to enhance model accuracy by iteratively correcting the errors of weak classifiers. At each iteration, it focuses on the misclassified samples from previous rounds

and trains a new model that better captures these challenging instances. By combining the strengths of multiple weak models, it produces a robust classifier with improved accuracy and generalization. Since Helios is essentially a rule learner, which also acts as a weak classifier, we combine SMPL with boosting to further enhance classification performance.

## 3 OVERVIEW

In this paper, we propose Helios, a framework for learning and adaptation of matching rules for incremental attack classes, and achieving continual in-network malicious traffic detection. First, we design a novel Supervised Mixture Prototypical Learning (SMPL) method to encapsulate the knowledge of known classes (including benign traffic and known attacks) into a set of prototypes. Each prototype corresponds to a centroid of its class in the traffic feature space. By calculating the similarity between the input sample and prototypes, Helios achieves multi-class classification of known classes and identification of unseen attacks. Second, Helios converts the learned prototypes into a set of range-based matching rules. Although the prototypes are trained on the control plane (i.e., GPU-based server), this enables inference on the data plane of network devices (e.g., programmable switches), meeting the demands of line-rate processing for high-speed traffic. Finally, after network administrators complete the collection and labeling of data from the unseen attack, it becomes a newly known attack, and the corresponding new samples are added to the training set. Helios supports incremental rule learning and updating to reduce training time and minimize packet interruption caused by hardware reconfiguration.

As illustrated in Figure 1, Helios consists of three modules: attack knowledge prototyping, priority-guided rule transformation, and class-incremental rule adaptation.

**Attack Knowledge Prototyping.** The attack knowledge prototyping module distills prototypes that encapsulate the knowledge of each known class. To improve inter-prototype discrimination, Helios leverages the density-based clustering method DBSCAN [12] for prototype initialization, ensuring that each class is assigned a number of prototypes proportional to its data complexity. Helios employs the weighted infinity norm distance as the similarity metric to facilitate the subsequent rule conversion. During SMPL, Helios uses gradient descent to increase the similarity between each prototype and the features of samples from the same class while reducing similarity with features of samples from different classes. After training, if a test sample exhibits low similarity to all prototypes, it is classified as unknown.

**Priority-Guided Rule Transformation.** The priority-guided rule transformation module first converts the trained prototypes into range-based matching rules. Specifically, Helios partitions the feature space by leveraging the sample-prototype associations and calibrates the boundaries to generate prototype-generated rules. To resolve overlaps between these rules, which may cause issues when multiple switch table entries are matched simultaneously, Helios calculates the priorities of these rules using topological sorting. Additionally, Helios introduces the overlapping regions that do not achieve optimal classification results as higher-priority rules. Finally, Helios iteratively performs boosting on misclassified residual

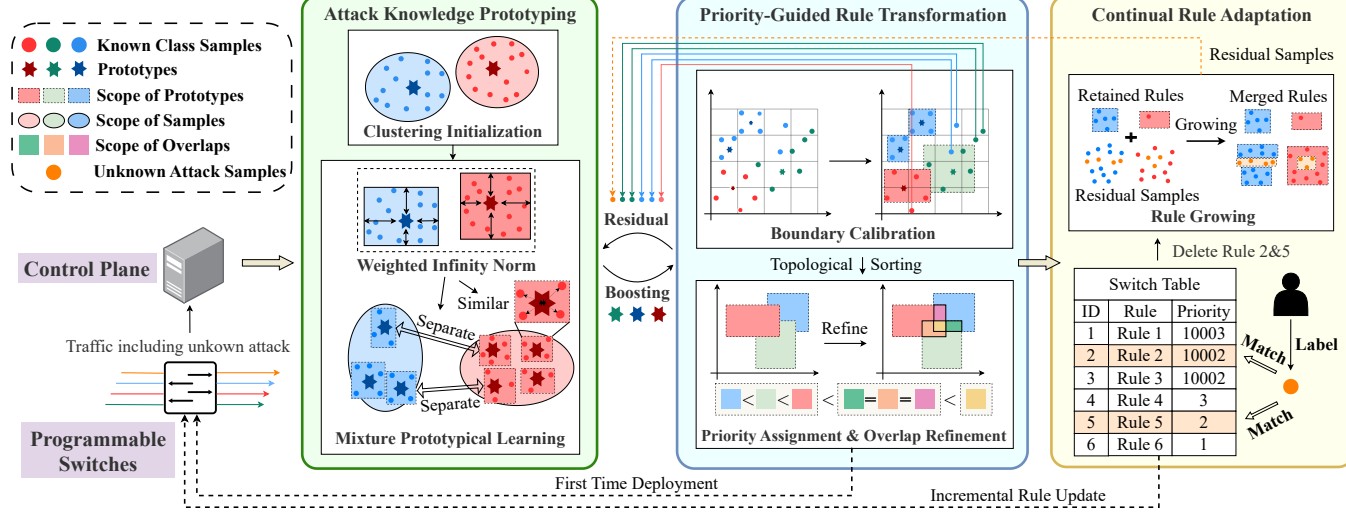

**Figure 1: The workflow of Helios.**

samples, re-prototyping them. After each iteration, the resulting rules are merged and all rule priorities are reassigned.

**Class-Incremental Rule Adaptation.** The class-incremental rule adaptation module performs incremental prototype learning and rule updates when a new attack class emerges (e.g., identified by the network administrator). Helios only makes necessary modifications to the existing rules, retaining those that are not matched by any new attack samples. On this basis, the new training set includes both new attack instances and existing misclassified residual samples. Similar to the boosting process, Helios then conducts incremental learning on the updated training set. The newly learned rules are incrementally incorporated, ensuring low-cost reconfiguration of hardware table entries.

## 4 METHODOLOGY

In this section, we present the details of Helios, including the attack knowledge prototyping module, the priority-guided rule transformation module, and the class-incremental rule adaptation module.

### 4.1 Attack Knowledge Prototyping

Existing prototypical learning methods typically assign only a single prototype per class. This is effective in GPU-based servers due to the integration of powerful feature extraction modules (e.g., deep neural networks) in an end-to-end learning manner, resulting in minimal overlap. However, programmable switches do not support extended feature transformations, as they involve floating-point operations and require complex computational logic. In Helios, we directly compare the raw features with prototypes and assign multiple prototypes to each class with an innovative supervised mixture prototypical learning (SMPL) method. Algorithm 1 illustrates the overall process.

Let $D = \{(x_i, y_i)\}_{i=1}^{N}$ represent the training dataset, where $x_i \in \mathbb{R}^d$ is the feature vector of the $i$-th sample, and $y_i \in C$ is the corresponding label from the set of known classes (including benign and known attacks). Due to the large range of feature values (e.g.,

0 to 65535), we apply min-max normalization to standardize them before inputting them into the model.

**Prototype Initialization.** Proper prototype initialization is critical to reflect data complexity. Insufficient prototypes fail to capture essential patterns, while excessive ones lead to overfitting. Additionally, careful initialization promotes efficient parameter convergence. Helios utilizes unsupervised clustering to initialize prototypes around cluster centers. Instead of using $K$-means [17], which requires specifying the number of clusters in advance, we adopt DBSCAN [12] for its ability to identify clusters based on data density. DBSCAN defines clusters using two parameters: the radius of neighborhoods, and the minimum number of points required to form a core cluster.

Specifically, for the $i$-th class, the initial value of the $j$-th prototype $P_{ij}$ is set as the average of the $j$-th cluster obtained by applying DBSCAN on the training samples labeled as class $i$: $P_{ij} = \frac{1}{|U_{ij}|} \sum_{x_k \in U_{ij}} x_k$, where $U_{ij}$ is the $j$-th cluster from DBSCAN on $\{x_k \mid y_k = i\}$. Initializing prototypes from cluster centers facilitates better convergence and enhances representational capacity.

**Supervised Mixture Prototypical Learning.** To measure the similarity between features and prototypes, a classic choice is the Euclidean distance. However, data planes do not efficiently support square computations. We adopt the infinity norm $\| \cdot \|_\infty$, which calculates the maximum difference between the feature vector and the prototype across all dimensions. This approach aligns with the nature of the upper and lower bounds in matching rules. Additionally, we introduce a feature weight parameter $w_{ij}$ for each prototype $P_{ij}$, which scales the differences across dimensions to handle feature values more flexibly. The initial values of $w$ are set to 1, which means the identical mapping. By taking the absolute value of original differences and weights, we ensure the final distance is positive. Consequently, the distance metric is defined as follows:

$$\text{Dist}(x, P_{ij}) = \left\| \left| \frac{x - x_{min}}{x_{max} - x_{min}} - P_{ij} \right| \cdot |w_{ij}| \right\|_\infty, \quad (1)$$

where $x_{min}$ and $x_{max}$ denote the minimum and maximum values of the features, respectively. $P_{ij}$ represents the $j$-th prototype of the $i$-th class, and $w_{ij}$ is the corresponding feature weight parameter.

To determine the classification probability distribution for a given sample, we first calculate the distance between the sample feature vector and all prototypes using distance metric (1). We then define the sample's distance to a specific class as the minimum distance to all prototypes within that class. The classification probability distribution is then obtained by applying the softmax function to these distances. Additionally, we introduce a temperature parameter $T$ to control the smoothness of the probability distribution [19]. It scales the distances before applying the softmax function, thereby enhancing the convergence of prototype training. Consequently, the classification probability for the $i$-th class is as follows:

$$\text{Prob}(y = i \mid x) = \frac{\exp\left(\min_j \text{Dist}(x, P_{ij})/T\right)}{\sum_k \exp\left(\min_j \text{Dist}(x, P_{kj})/T\right)}. \quad (2)$$

We use the cross-entropy loss to simultaneously train prototypes $P$ and weight parameters $w$ via gradient descent. Therefore, in each iteration of training for a sample, only the nearest prototype of each class participates in the probability calculation and undergoes corresponding gradient updates. Among these prototypes, the one with the same label as the sample is pulled closer, while those from other classes are pushed away. This ensures that prototypes unrelated to the current training sample remain unaffected.

## 4.2 Priority-Guided Rule Transformation

The trained prototypes need to be deployed on the data plane to assign labels for incoming traffic samples. The basic idea is to compare the input sample with the prototypes of all classes and find the nearest one according to the distance metric. If the minimum distance falls below the acceptance threshold, the sample is assigned to the corresponding class of this prototype. Otherwise, it is identified as a new attack type. However, directly deploying this inference process on switches is impractical since distance calculations and comparisons are difficult to implement within the limited stages of the switch pipeline. To address this, Helios design a priority-guided method that transforms the inference process of prototypes into range-based rule matching, making it more suitable for deployment. Algorithm 2 illustrates the overall process of rule transformation.

**Boundary Calibration.** Since each sample is accepted by its nearest prototype, it is essential to set acceptance thresholds for each prototype to identify unseen attacks. One straightforward approach is to set the threshold as the maximum distance among all accepted samples. However, overly large acceptance boundaries can reduce the generalization ability. We observe that most accepted samples are relatively close to their prototypes in practice, with only a few outliers. Therefore, in Helios, we define the threshold as the mean distance of the accepted training samples, resulting in a tighter boundary. This approach provides a robust threshold by ensuring that the acceptance boundary is dominated by closer samples, thereby preventing it from being influenced by a few distant outliers. The corresponding formulas are given below:

$$D_{P_{ij}} = \{x_k \mid y_k = i \wedge \text{Dist}(x_k, P_{ij}) \leq \text{Dist}(x_k, P_{i'j'}), \forall P_{i'j'}\}, \quad (3)$$

$$\text{Threshold}(P_{ij}) = \frac{1}{|D_{P_{ij}}|} \sum_{x_k \in D_{P_{ij}}} \text{Dist}(x_k, P_{ij}), \quad (4)$$

where $D_{P_{ij}}$ represents the set of samples accepted by $P_{ij}$.

**Rule Transformation.** After calibrating the acceptance boundaries, Helios transforms the prototype inference process into range-based matching rules, which are directly supported by programmable switches. For a range rule with $d$ dimensions, an input sample is considered a match if it falls within the specified bounds for each dimension. If the sample exceeds the bounds in any dimension, it is treated as a miss. The formal expression is given below:

$$\text{Match}(x_i, l, u) = \bigwedge_{v=1}^{d} (l_v \leq x_{iv} \leq u_v), \quad (5)$$

where $l_v$ and $u_v$ represent the lower and upper bounds for dimension $v$, and $x_{iv}$ is the $v$-th dimension of the input sample $x_i$.

Using the acceptance relationship between samples and prototypes, Helios partitions the feature space and transforms each prototype into a corresponding rule. Specifically, the bounds for each feature dimension in the rule are determined by the minimum and maximum values of all accepted samples within the prototype's acceptance threshold. Formally, for the $v$-th feature dimension, the bounds of the rule corresponding to prototype $P_{ij}$ are defined as:

$$\{l_v, u_v\} = \left\{ \min_{x_r \in D_{P_{ij}}} x_{rv}, \max_{x_s \in D_{P_{ij}}} x_{sv} \right\}, \quad v = 1, 2, \ldots, d, \quad (6)$$

where $l$ and $u$ represent the lower and upper bounds, respectively.

**Overlap Refinement and Priority Assignment.** Since SMPL generates multiple prototypes, overlap issues inevitably arise. When rules overlap, their classification results may conflict, and the optimal class for an overlapping region may not align with any of the original rules that generated it. Given that overlapping regions represent a finer partition of the feature space, refining these regions can further improve performance. Ideally, each region should have an optimal classification result, defined as the class containing the largest number of samples within that region. Fortunately, programmable switches support a priority mechanism that can resolve conflicts by returning the result of the highest priority rule. Therefore, to mitigate these conflicts and maximize classification performance, Helios prioritizes the rules generated by prototypes and introduces additional higher-priority rules to cover the remaining overlapping regions that still do not achieve optimal classification.

First, all samples in the training set are matched against the rules to generate conflict sets. Each conflict set $S_i$ consists of a group of overlapping rules $\{R_{i1}, R_{i2}, \ldots, R_{im}\}$, where $R_{ij}$ denotes the $j$-th rule in the $i$-th conflict set. The class $C_i$ with the highest number of samples in the overlapping region is selected as the representative class for $S_i$. Next, a directed acyclic graph (DAG) $G$ is constructed. The conflict sets are then sorted in ascending order based on their sizes, and each conflict set is processed sequentially. For each $S_i$, directed edges $(R_{ij} \rightarrow R_{ik})$ are inserted to $G$ from all rules $R_{ij} \in S_i$ belonging to class $C_i$ to all other rules $R_{ik} \in S_i$ that do not belong to class $C_i$. If a path already exists from $R_{ik}$ to $R_{ij}$, the insertion of this edge is skipped. Finally, the priorities of the prototype-generated rules are assigned in descending order based on the results of topological sorting on $G$, starting from 1. For any conflict set $S_k$ that

remains unresolved, we introduce the corresponding overlapping regions as additional rules. These rules are assigned a priority of $10000 + |S_k|$, where $|S_k|$ denotes the size of conflict set $S_k$. This ensures that their priority surpasses all prototype-generated rules, allowing for complete coverage. Additionally, we observe that some rules accept only a small number of samples, making them less cost-effective. We prune these rules to mitigate overfitting, which also reduces the switch's resource consumption.

**Rule Boosting.** After completing the initial round of prototype training and rule transformation (including refinement), Helios utilizes the misclassified residual training samples based on the current rules to perform rule boosting. New prototypes are trained on these residual samples, and the resulting rules are merged into the existing rule set. Following this, Helios applies overlap refinement and priority assignment to the combined set of rules and then iteratively proceeds to the next round of boosting. Recall that Helios employs tight boundary calibration, which, in combination with boosting, maximizes the fit to the sample distribution and enhances overall classification performance.

### 4.3 Class-Incremental Rule Adaptation

For the initial classes, Helios is trained and deployed for the first time. When an unseen attack emerges, if a sample does not match any of the rules, it is identified as an unseen attack class. The classification results from the data plane provide network administrators with timely feedback and alerts. After network administrators collect and label new attack samples, Helios performs incremental learning on the updated dataset, enabling efficient rule updates and lightweight switch reconfiguration. Algorithm 3 illustrates the overall process of rule adaptation.

**Isolated Rule Retention.** During rule updates, a totally incremental approach would retain all existing rules while adding new ones learned from the new attack. However, this may lead to significant conflicts between merged rules, reducing learning capability and deployment efficiency. Thus, existing rules need to be adjusted necessarily, such as modifying their boundaries or splitting them [23]. Specifically, Helios retains those isolated rules that are not matched by any new attack samples in the training set, that is,

$$R_i \text{ is retained if } \forall x_k \in D_{\text{new attack}}, \neg \text{Match}(x_k, R_i). \quad (7)$$

By retaining rules that are unaffected by new attack samples, Helios achieves the trade-off between minimizing unnecessary changes and maintaining flexibility for updates.

**Incremental Rule Update.** After retaining the existing isolated rules, Helios performs incremental learning on the new dataset, which includes misclassified residual samples and new attack samples. Similar to the boosting-based SMPL, new rules are generated and incrementally merged into the existing set. Helios then applies overlap refinement and priority assignment across all rules. For the newly added rules, only incremental deployment to the switch is required, ensuring efficient reconfiguration. For retained prototype-generated rules, some may need priority adjustments based on the updated topological sorting results, unless they do not conflict with any new rules and thus remain unchanged. Additionally, if the optimal classification results change, corresponding modifications are also necessary. Compared to adding or deleting rules, modifying priorities or updating classification results is more efficient.

## 5 EXPERIMENTS

### 5.1 Settings

**Datasets.** (1) *CICIDS2018* [29], which includes network traffic generated within a simulated enterprise environment, featuring various attacks such as Distributed Denial of Service (DDoS) attacks (including LOIC and HOIC methods), Denial of Service (DoS) attacks (e.g., GoldenEye, Hulk, Slowloris), and SSH brute force attempts. (2) *TON-IoT* [3], specifically designed for Internet of Things (IoT) applications, incorporating benign traffic alongside nine distinct attack scenarios. (3) *UNSW-NB15* [28], which integrates real normal activities with synthetic contemporary attack behaviors, featuring benign traffic and encompassing nine types of attacks.

**Baselines.** Given that methods supporting only binary classification are unsuitable for class-incremental learning scenarios, we select four state-of-the-art multi-classification in-network methods as baselines: 1) *IIsy* [38], which designs a feature encoding approach for deploying decision trees; 2) *Planter* [41] and *Netbeacon* [42], which utilize different ensemble encoding methods for offloading random forests; and 3) *Mousikav2* [37], a lightweight method based on knowledge distillation.

**Metrics.** We evaluate the multi-classification performance for known classes using accuracy (ACC) and F1-score (F1). For the identification of unseen attacks, we calculate the True Positive Rate (TPR) and the Area Under the Curve (AUC) under various threshold configurations to provide a more comprehensive evaluation.

**Configurations.** We divide the training and updating process into several tasks, each consisting of classification for the currently known classes and the identification of an unseen attack. We set the training epochs for each task to 50 and utilize the Adam [20] optimizer with a learning rate of 0.001. The default values for other key hyper-parameters are provided in Section A. Considering the limited well-labeled samples in the real-world environment, we set a 2:8 ratio of training set and testing set to simulate few-shot learning scenarios. Additionally, since the baselines do not inherently support unknown class identification, we extend them in a manner similar to previous out-of-distribution detection arts [18, 24, 33, 35] by employing a threshold on the classification probabilities. Specifically, if the classification probabilities for all classes fall below the threshold, the input sample is classified as unknown. We calculate the optimal threshold for each baseline by maximizing 5×ACC+TPR while ensuring that the ACC remains no lower than 85%.

### 5.2 Classification Performance Evaluation

We compare the classification performance of Helios with enhanced baseline methods across three datasets, as shown in Table 1. The tasks are divided based on the number of classes in each dataset, with the specific attack details in Table 3. Initially (Task 1), the model classifies the initial classes (including benign and one attack) and identifies the first unseen attack. In each subsequent task, the unseen attack from its previous task is added as known, while a new attack is introduced as unseen for that task. Finally (Task ALL), ACC and F1 represent performance across all classes, while TPR is calculated as the weighted average of all previous tasks, representing the overall identification rate for unseen attacks.

**Table 1: Comparisons of Helios with prior arts on each task in terms of classification performance. Here, ACC, F1, and TPR represent the Accuracy (%), F1-score (%) of known class classification, and the True Positive Rate (%) of new attack identification.**

| Dataset | Task | Planter | | | Netbeacon | | | Mousikav2 | | | IIsy | | | Helios | | |
|---|---|---|---|---|---|---|---|---|---|---|---|---|---|---|---|---|
| | | ACC | F1 | TPR | ACC | F1 | TPR | ACC | F1 | TPR | ACC | F1 | TPR | ACC | F1 | TPR |
| IDS | 1 | 100.00 | 100.00 | 0.00 | 100.00 | 100.00 | 0.00 | 98.86 | 99.39 | 75.85 | 99.98 | 99.98 | 0.00 | 99.18 | 99.59 | 100.00 |
| | 2 | 99.95 | 99.95 | 0.00 | 99.95 | 99.95 | 0.00 | 99.80 | 99.80 | 0.00 | 99.97 | 99.97 | 0.00 | 99.27 | 99.63 | 100.00 |
| | 3 | 99.91 | 99.95 | 98.30 | 99.91 | 99.95 | 98.30 | 99.88 | 99.91 | 35.10 | 99.96 | 99.96 | 0.00 | 99.33 | 99.66 | 100.00 |
| | 4 | 99.74 | 99.86 | 22.85 | 99.74 | 99.86 | 22.85 | 99.83 | 99.83 | 0.00 | 99.94 | 99.94 | 0.00 | 98.84 | 99.41 | 56.54 |
| | 5 | 90.76 | 93.73 | 59.10 | 88.87 | 92.40 | 59.10 | 91.50 | 91.50 | 0.00 | 94.89 | 94.83 | 0.00 | 92.50 | 92.86 | 100.00 |
| | 6 | 95.43 | 95.37 | 0.00 | 94.59 | 94.53 | 0.00 | 92.74 | 92.74 | 0.00 | 95.52 | 95.46 | 0.00 | 93.73 | 94.08 | 100.00 |
| | 7 | 89.05 | 92.85 | 100.00 | 88.99 | 92.72 | 100.00 | 93.73 | 93.73 | 0.00 | 95.35 | 95.30 | 0.00 | 94.69 | 95.01 | 100.00 |
| | 8 | 89.90 | 93.44 | 92.31 | 89.67 | 92.96 | 82.05 | 93.59 | 93.60 | 10.26 | 95.35 | 95.30 | 0.00 | 95.85 | 96.04 | 100.00 |
| | 9 | 90.47 | 93.81 | 98.47 | 94.57 | 94.40 | 97.71 | 86.12 | 89.75 | 99.24 | 95.33 | 95.28 | 0.00 | 95.79 | 95.98 | 100.00 |
| | ALL | 92.11 | 94.71 | 26.78 | 94.25 | 94.06 | 26.76 | 93.54 | 93.45 | 33.21 | 95.27 | 95.23 | 0.00 | 95.78 | 95.98 | 98.21 |
| IoT | 1 | 99.91 | 99.95 | 50.22 | 99.91 | 99.95 | 50.22 | 97.20 | 95.83 | 0.00 | 100.00 | 100.00 | 0.00 | 95.77 | 96.48 | 100.00 |
| | 2 | 96.04 | 97.47 | 92.05 | 95.93 | 97.37 | 91.95 | 95.84 | 96.46 | 11.79 | 98.48 | 98.53 | 0.00 | 94.96 | 96.56 | 100.00 |
| | 3 | 98.89 | 98.89 | 98.79 | 97.41 | 98.19 | 98.83 | 94.28 | 94.76 | 58.75 | 98.92 | 98.89 | 0.00 | 95.52 | 96.91 | 99.16 |
| | 4 | 97.83 | 98.40 | 60.41 | 97.68 | 98.35 | 99.01 | 97.09 | 97.55 | 43.83 | 98.87 | 98.87 | 0.11 | 97.05 | 97.97 | 23.63 |
| | 5 | 98.73 | 99.10 | 89.19 | 98.61 | 98.99 | 89.63 | 99.03 | 99.07 | 4.87 | 99.29 | 99.28 | 0.71 | 98.11 | 98.47 | 92.62 |
| | 6 | 96.47 | 97.69 | 39.01 | 92.89 | 96.04 | 56.07 | 97.11 | 97.23 | 7.27 | 97.43 | 98.19 | 12.74 | 96.26 | 97.18 | 45.36 |
| | 7 | 85.59 | 91.16 | 99.96 | 85.30 | 90.58 | 99.85 | 93.42 | 93.36 | 0.00 | 97.91 | 97.90 | 0.00 | 95.74 | 96.34 | 99.80 |
| | 8 | 88.98 | 92.89 | 54.99 | 91.94 | 94.02 | 40.54 | 91.25 | 93.64 | 31.28 | 96.61 | 97.78 | 25.95 | 97.04 | 97.68 | 25.84 |
| | ALL | 88.60 | 88.43 | 71.83 | 86.58 | 86.53 | 79.06 | 83.01 | 87.01 | 21.19 | 91.72 | 91.72 | 6.87 | 92.20 | 92.99 | 62.93 |
| NB15 | 1 | 97.38 | 98.57 | 55.59 | 98.05 | 98.48 | 73.84 | 95.37 | 97.08 | 71.79 | 99.10 | 99.10 | 0.00 | 94.42 | 96.96 | 99.09 |
| | 2 | 96.35 | 97.95 | 21.30 | 98.09 | 98.19 | 1.94 | 88.10 | 89.71 | 38.31 | 99.35 | 99.34 | 0.00 | 91.86 | 95.20 | 63.13 |
| | 3 | 91.80 | 95.24 | 57.35 | 95.30 | 95.16 | 0.00 | 87.90 | 90.89 | 17.15 | 97.76 | 97.75 | 0.00 | 90.68 | 94.03 | 86.14 |
| | 4 | 88.21 | 92.78 | 58.77 | 88.18 | 87.67 | 0.00 | 90.40 | 91.37 | 3.57 | 96.78 | 96.77 | 0.00 | 90.51 | 93.87 | 55.35 |
| | 5 | 85.78 | 91.44 | 87.68 | 79.61 | 77.02 | 0.00 | 85.07 | 85.21 | 0.16 | 95.29 | 95.28 | 0.00 | 86.37 | 89.92 | 73.19 |
| | 6 | 86.08 | 91.58 | 90.77 | 80.30 | 78.33 | 0.00 | 85.71 | 86.50 | 1.78 | 95.54 | 95.54 | 0.00 | 89.43 | 92.65 | 66.59 |
| | 7 | 91.00 | 93.08 | 27.78 | 69.30 | 67.74 | 0.00 | 82.14 | 82.24 | 0.00 | 93.07 | 93.04 | 0.00 | 89.44 | 92.50 | 62.43 |
| | 8 | 91.12 | 92.28 | 45.71 | 73.74 | 72.20 | 0.00 | 83.84 | 83.22 | 0.00 | 92.08 | 92.08 | 0.00 | 91.53 | 93.11 | 69.30 |
| | ALL | 81.79 | 88.79 | 62.65 | 75.64 | 73.55 | 1.22 | 79.88 | 81.60 | 6.02 | 91.83 | 91.83 | 0.00 | 92.91 | 94.19 | 69.15 |

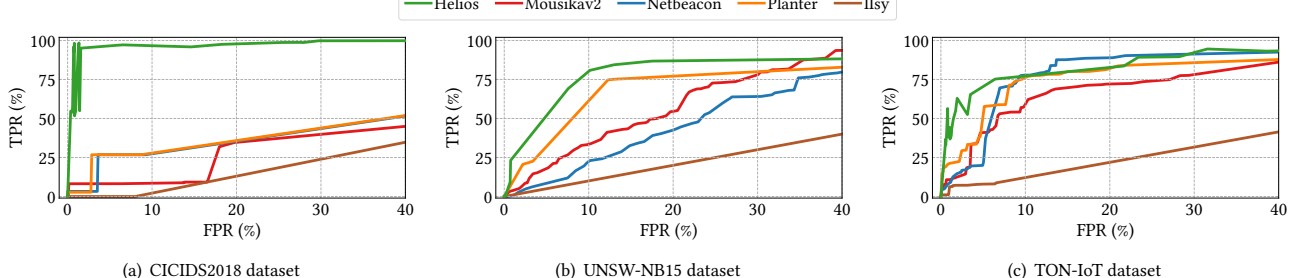

(a) CICIDS2018 dataset  (b) UNSW-NB15 dataset  (c) TON-IoT dataset

**Figure 2: Receiver Operating Characteristic (ROC) curve of new attacks identification on three datasets.**

For overall performance in Task ALL, Helios achieves the highest ACC and F1 across all datasets, as well as the best TPR on CICIDS2018 and UNSW-NB15, demonstrating superior classification precision. For individual tasks, Helios shows more stable performance compared to the baselines, highlighting its ability to handle continually emerging unseen attacks. In summary, even after optimizing the existing state-of-the-art in-network methods and selecting their optimal thresholds, Helios consistently outperforms them in classifying known classes and identifying unseen attacks.

To further demonstrate the model's capability in identifying unseen attacks under different threshold settings, we plot ROC curves, as shown in Figure 2. Across all datasets, Helios outperforms the baseline methods, achieving higher TPR while maintaining lower FPR. On CICIDS2018, Helios achieves an AUC of 0.98, significantly outperforming the second-best method, Planter, which achieves 0.59. On TON-IoT and UNSW-NB15, Helios also surpasses its top

competitors, with AUC of 0.90 and 0.88, compared to 0.88 for Netbeacon on TON-IoT and 0.81 for Planter on UNSW-NB15, respectively.

## 5.3 Reconfiguration Time Evaluation

We conduct training on servers equipped with Intel (R) Xeon (R) Silver 4210 CPU @ 2.20GHz and V100 GPUs, and deploy the model on a commodity Tofino switch (Edgecore Wedge100BF-65X[3]). Figure 3 presents the average switch table reconfiguration time for different methods, along with the number of rules learned on total classes. Overall, Helios outperforms the most lightweight but less accurate method, Mousikav2, while significantly surpassing other methods. As shown in Figure 3(a), Helios achieves minimal rule update time overhead, ranging from 0.8 to 3.1 seconds. This result can be attributed to the number of rules learned, as depicted in Figure 3(b), where Helios demonstrates higher learning efficiency by requiring fewer rules. Additionally, due to its incremental update mechanism,

---

[3]https://www.edge-core.com/product/dcs802/

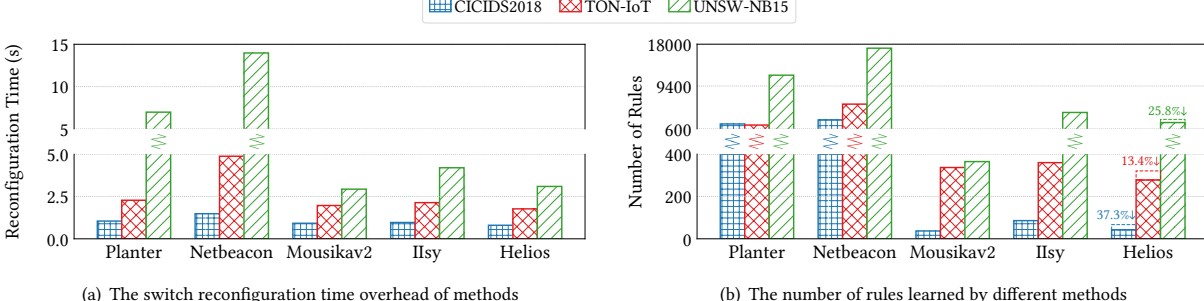

(a) The switch reconfiguration time overhead of methods

(b) The number of rules learned by different methods

**Figure 3: Comparison of switch reconfiguration time and the number of learned rules between Helios and the baseline methods across three datasets. For Helios, the dashed portion of the bar represents the savings during incremental updates.**

Helios achieves a 13.4% to 37.3% reduction in rule deployment, further reducing time overhead. In conclusion, Helios enables efficient switch reconfiguration and minimizes switch interruption time.

## 5.4 Ablation Study

To further validate the design of Helios, we conducted ablation studies on the TON-IoT dataset to assess the contribution of each module, as detailed in Table 2. First, initializing prototypes with DBSCAN yields significant improvements in both ACC and TPR compared to Normal or Uniform initialization, and it also surpasses $K$-means initialization (we set sufficient clusters for $K$-means to ensure consistency). Second, using Euclidean distance decreases ACC, suggesting that $l_\infty$ distance provides a better delineation of decision boundaries and is more suitable for conversion to range-based rules. Next, initializing prototypes without subsequent boosting, or performing only a single training iteration after initialization, fails to achieve high ACC. While continual boosting without SMPL improves ACC, it significantly lowers TPR. These results demonstrate that both SMPL and boosting are effective methods for enhancing performance. Finally, without boundary calibration, although slightly increasing ACC, results in a notable decrease in TPR. This highlights the importance of refining the accepting boundary of prototypes. Overall, Helios achieves an optimal balance between ACC and TPR, ensuring high precision in classifying known classes while effectively identifying unknown attacks.

**Table 2: The ablation study of key components in Helios, where NOR represents the number of rules.**

| Method | ACC(%) | TPR(%) | NOR |
|---|---|---|---|
| w/ Normal init. | 76.91 | 36.38 | 90 |
| w/ Uniform init. | 82.00 | 33.18 | 61 |
| w/ $K$-means | 91.46 | 58.30 | 394 |
| w/ Euclidean dist. | 90.08 | 64.66 | 306 |
| w/o SMPL and Boosting | 66.08 | 87.76 | 161 |
| w/o Boosting | 70.39 | 78.18 | 146 |
| w/o SMPL | 88.52 | 38.39 | 325 |
| w/o Boundary Calibration | 92.48 | 38.91 | 528 |
| Helios | 92.20 | 62.93 | 321 |

## 5.5 Hardware Performance

The hardware performance of Helios is depicted in Figure 4. Figure 4(a) illustrates the memory consumption, showing that Helios typically consumes less than 10% of TCAM and SRAM. Even for the most complex dataset, UNSW-NB15, Helios requires only around 40% of TCAM. This efficiency ensures sufficient resources remain available for other essential network functions, such as routing. Additionally, we use a traffic generator (SPIRENT N11U[4]) to simulate high-speed network traffic at 10 Gbps, 50 Gbps, and 100 Gbps, with the throughput and latency results presented in Figure 4(b). As shown, Helios achieves high-speed processing without packet loss, while maintaining notably low latency (around 0.66 $\mu s$) across varying input traffic rates. Therefore, Helios enables high-throughput, low-latency detection of malicious traffic.

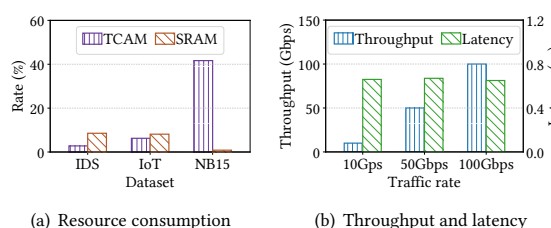

(a) Resource consumption

(b) Throughput and latency

**Figure 4: Hardware performance of Helios.**

## 6 CONCLUSION

In this paper, we propose Helios, a continual in-network malicious traffic detection framework for attack-incremental scenarios. Specifically, Helios integrates supervised mixture prototypical learning with boosting to derive prototypes that represent the knowledge of each class, facilitating the classification of known classes and the identification of unknown attacks. The inference process of prototypes is then transformed into priority-based rule matching, ensuring accurate and efficient switch deployment. Helios also supports incremental prototype learning and rule updates when new attacks are incorporated, achieving low-cost hardware reconfiguration. Extensive evaluations of Helios using three datasets demonstrate its effectiveness in identifying unknown attacks and performing efficient updates.

[4]https://support.spirent.com/SpirentCSC/SC_KnowledgeView?Id=DOC10479

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

## A   PARAMETERS ANALYSIS

We analyze the impact of three key parameters on the performance of Helios: boosting iteration, pruning lower bound, and calibration threshold. The experimental results are presented in Figure 5.

**Boosting iteration.** As shown in Figure 5(a), increasing boosting iterations initially improves ACC as the model handles misclassified samples better. However, beyond a certain point, both ACC and TPR reach a state of convergence, indicating sufficient learning. To prevent overfitting and excessive rule generation, the default boosting iteration is set to 4, balancing accuracy and model complexity.

**Pruning lower bound.** Figure 5(b) shows that without pruning, both ACC and TPR are lower compared to when a smaller pruning lower bound (e.g., 10) is applied. This is because overfitted rules from the training set reduce generalization on the test set and occupy unnecessary feature space. As the pruning lower bound increases, rule complexity decreases, but ACC also declines. Therefore, a smaller pruning lower bound provides better results, and the default value is set to 10.

**Calibration threshold.** Recall that we use the mean distance of a prototype's accepted samples as the threshold (Section 4.2), and here we multiply this threshold by a scaling weight for parameter analysis. As shown in Figure 5(c), when the scaling weight is below 1.0, ACC increases significantly because smaller thresholds fail to capture all relevant data. When the scaling weight exceeds 1.0, ACC stops improving, while TPR decreases due to overly broad rules influenced by noisy samples. Therefore, consistent with Eq.(4), the default value is set to 1.0.

## B   MATCHING TABLE IMPLEMENTATION

As illustrated in Listing 1, features are used as keys for the matching table, and table entries are processed sequentially according to their assigned priorities. When a match is found, the corresponding `Set_class` action is executed, classifying the sample as either a known attack or benign traffic. If no table entry matches, the `Set_as_unseen_attack` action is triggered, classifying the sample as an unseen attack. This mechanism ensures that new or unknown traffic is properly flagged for further analysis in the control plane.

### Listing 1: The P4 matching table.

```
1  table Helios {
2      key = {
3          meta.feature_1: range; // Range-based matching
4          meta.feature_2: range;
5          ...
6          meta.feature_m: range;
7      }
8      actions = {Set_class_1; ..., Set_class_n;} // Hit
9      default_action = {Set_as_unseen_attack;} // Miss
10  }
```

## C   PSEUDOCODE OF ALGORITHMS

We present the pseudocode for various modules of Helios, including the attack knowledge prototyping module (Algorithm 1), the priority-guided rule transformation module (Algorithm 2), and the class-incremental rule adaptation module (Algorithm 3). These modules collectively enable Helios to achieve high classification performance while maintaining efficient switch reconfiguration.

---

**Algorithm 1:** Attack Knowledge Prototyping

**Input:** Training set $D = \{(x_i, y_i)\}_{i=1}^N$
**Output:** Prototypes P, Weight parameters w
1  Run DBSCAN on $D$ to initialize $P$ as cluster centroids;
2  $w \leftarrow 1$;
3  **for** *each epoch* **do**
4      **for** *each mini-batch* $\{(x_k, y_k)\}$ *in D* **do**
5          $\hat{y}_k \leftarrow \text{Prob}(x_k)$ via Eq.(2);
6          $L \leftarrow \text{Loss}(\hat{y}_k, y_k)$;
7          Update $P$ and $w$ using gradients $\nabla_P L$, $\nabla_w L$;
8      **end**
9  **end**
10  **return** $P$, $w$;

---

**Algorithm 2:** Priority-Guided Rule Transformation

**Input:** Training set $D = \{(x_i, y_i)\}_{i=1}^N$
**Output:** Range-based matching rules $R$
1  Initialize prototype-generated rules $R_{gen}$ as empty;
2  $D_{now} \leftarrow D$;
3  **for** *each boosting iteration* **do**
4      $P \leftarrow$ Perform Algorithm 1 on $D_{now}$;
5      **for** $P_{ij}$ *in P* **do**
6          Compute $D_{P_{ij}}$ via Eq.(3);
7          Compute Threshold$_{ij}$ via Eq.(4);
8          Compute boundaries $(L_{ij}, U_{ij})$ for $P_{ij}$ via Eq.(6);
9          Add $(L_{ij}, U_{ij})$ to $R_{gen}$;
10      **end**
11      Refine the optimal class for all overlapping regions in $R_{gen}$;
12      Perform topological sorting on $R_{gen}$ and assign priorities;
13      $R_{overlap} \leftarrow$ Introduce regions still not achieve optimal;
14      Prune low cost-effective rules in $R_{gen}$ and $R_{overlap}$;
15      $D_{now} \leftarrow$ Residual samples of $D_{now}$ based on $R_{gen}$ and $R_{overlap}$;
16  **end**
17  $R_{overlap} \leftarrow$ Overlap refinement and priority assignment on $R_{gen}$;
18  $R \leftarrow R_{gen} \cup R_{overlap}$;
19  **return** $R$;

---

**Algorithm 3:** Class-Incremental Rule Adaptation

**Input:** Known class samples $D_{known}$, new attack samples $D_{new}$, rules learned on known class samples $R_{exist}$
1  $D_{update} \leftarrow D_{known} \cup D_{new}$;
2  $R_{isolate} \leftarrow R_{exist} \setminus \{\text{rules that match any } x_k \in D_{new}\}$;
3  $D_{residual} \leftarrow \{x \in D_{update} \mid x \text{ is misclassified by } R_{isolate}\}$;
4  $R_{new} \leftarrow$ Perform Algorithm 2($D_{residual}, R_{exist}$);
5  $R_{retain} \leftarrow R_{exist} \cap R_{new}$;
6  $R_{modify} \leftarrow \{r \in R_{retain} \mid \text{priority}(r) \neq \text{priority}(R_{new})\}$;
7  $R_{delete} \leftarrow R \setminus R_{retain}$;
8  $R_{add} \leftarrow R_{new} \setminus R_{retain}$;
9  Perform the corresponding switch table-entry reconfiguration for $R_{modify}$, $R_{delete}$ and $R_{add}$;
10  **return**;

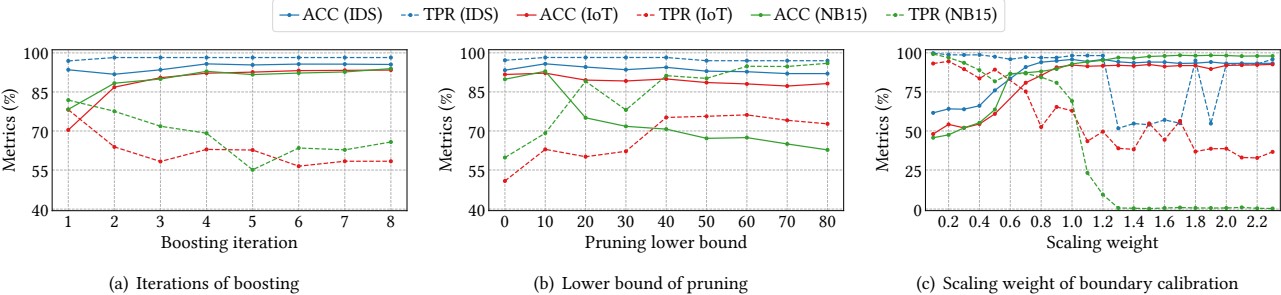

Figure 5: Analysis of key hyper-parameters for Helios.

## D  DATASET DETAILS

Table 3 presents the attack classes assigned to each task across datasets. Initially, each dataset begins with benign traffic and a single attack. As tasks progress, additional attack classes are introduced incrementally, simulating a continually evolving network environment. This incremental setup effectively evaluates the methods' ability to handle both known classes and newly emerging threats, reflecting the attack-incremental nature of real-world scenarios.

Table 4 presents the extracted traffic features used in our experiments. For UNSW-NB15, we extract features across various levels, including IPv4 (e.g., length, flags, TTL, protocol, and ports), TCP (e.g., offset, flags, and window size), and UDP (e.g., length). If a packet is of the TCP type, the UDP fields are padded with zeros, and vice versa. For TON-IoT, features such as total packet size and inter-arrival time are considered, with average, maximum, and minimum values captured to characterize the flow. Additionally, packet-level attributes such as packet count, protocol, and destination port are included. For CICIDS2018, flow-level features such as forward and backward packet sizes are extracted, along with packet-level attributes.

Table 3: Task-specific class details for each dataset.

| Task | CICIDS2018 | TON-IoT | UNSW-NB15 |
|---|---|---|---|
| Init | Benign | Benign | Benign |
| Init | DDoS LOIC HTTP | Mitm | Analysis |
| 1 | DDoS HOIC | DoS | Worms |
| 2 | DDoS LOIC UDP | Runsomware | Backdoor |
| 3 | DoS GoldenEye | Backdoor | DoS |
| 4 | DoS Hulk | Injection | Exploits |
| 5 | DoS Slowloris | DDoS | Fuzzers |
| 6 | SSH BruteForce | Password | Generic |
| 7 | Web Attack XSS | Scanning | Reconnaissance |
| 8 | Web Attack SQL | XSS | Shellcode |
| 9 | Web Attack Brute Force | - | - |

Table 4: Detailed features extracted from different fields, including both packet-level and flow-level features.

| Dataset | Field | Features |
|---|---|---|
| **NB15** | IPv4 | length, flags, TTL, protocol, srcport, dstport |
| | TCP | offset, flags, window_size |
| | UDP | length |
| **IoT** | Total Packet Size | avg, max, min |
| | Inter-Arrival Time | avg, max, min |
| | Packet-Level | pkt_count, protocol, dstport |
| **IDS** | Forward Packet Size | avg, max, min |
| | Backward Packet Size | avg, max, min |
| | Packet-Level | pkt_count, protocol |

