# OpenReview forum: "Helios: Learning and Adaptation of Matching Rules for Continual In-Network Malicious Traffic Detection"
_ACM.org/TheWebConf/2025/Conference — WWW 2025 Poster_

### Official Review · Reviewer_WFqz · 2024-11-25

**Novelty:** 5
**Technical Quality:** 4

**Review:**

This paper proposes a continuous in-network malicious traffic detection framework called **Helios** to address the limitations of traditional Network Intrusion Detection Systems (NIDS) in classifying known attacks and detecting unknown attacks. Experiments evaluated on three datasets (CICIDS2018, TON-IoT, and UNSW-NB15) show that Helios outperforms existing methods in terms of classification accuracy (ACC), F1 score, and unknown attack detection rate (TPR). At the same time, Helios demonstrates efficiency in hardware resource utilization and reconfiguration time. Well written.

**Questions:**

1. The **Helios** method is divided into multi-step solutions, such as DAG, and DBSCan, to analyze the complexity of Helios in detail.
2. Compare some incremental learning traffic intrusion detection algorithms. For example, T-DFNN: An Incremental Learning Algorithm for Intrusion Detection Systems.

**Reviewer Confidence:**

2: The reviewer is willing to defend the evaluation, but it is likely that the reviewer did not understand parts of the paper

**Scope:**

3: The work is somewhat relevant to the Web and to the track, and is of narrow interest to a sub-community

---

### Official Review · Reviewer_SFwM · 2024-11-27

**Novelty:** 5
**Technical Quality:** 6

**Review:**

This paper proposes a novel network intrusion detection system, Helios, to address the challenge of continuous malicious traffic detection in incremental attack scenarios. Overall, the paper is well-written, logically clear, and innovative. It demonstrates the superior performance of the proposed model by comparing it with other state-of-the-art baseline methods in three key areas: proficient classification ability, scalable hardware deployment, and incremental model updates.

Strengths:

- The paper introduces an innovative supervised hybrid prototype learning (SMPL) method that combines clustering initialization, which enhances the model’s performance.
- Experimental results show that Helios excels in both known class classification and unknown attack detection.

**Questions:**

I have no further questions regarding this paper.

**Reviewer Confidence:**

1: The reviewer's evaluation is an educated guess

**Scope:**

3: The work is somewhat relevant to the Web and to the track, and is of narrow interest to a sub-community

---

### Official Review · Reviewer_Phqm · 2024-12-02

**Novelty:** 5
**Technical Quality:** 5

**Review:**

## Summary
In this paper, the authors propose Helios, an in-network malicious traffic detection system designed for continual adaptation in attack-incremental scenarios. Helios leverages a Supervised Mixture Prototypical Learning (SMPL) method combined with clustering initialisation to classify known attacks and identify unseen ones.

## Strength
The paper is well-structured and provides a comprehensive overview of the problem, proposed solution, and experimental results. The methodology is clearly explained, and the authors used real-world datasets for evaluation.

## Weaknesses
1. Some sections could benefit from more detailed explanations, particularly the technical aspects of the SMPL method and the incremental update mechanism.
2. While the combination of techniques is novel, the individual components (e.g., prototypical learning, clustering) are well-established in the literature.

**Questions:**

1. Can you elaborate on the potential challenges and solutions for integrating Helios with existing network infrastructure?
2. How does the system ensure the security and integrity of the incremental updates?

**Reviewer Confidence:**

1: The reviewer's evaluation is an educated guess

**Scope:**

3: The work is somewhat relevant to the Web and to the track, and is of narrow interest to a sub-community